# Towards a Better Understanding of the Factors Associated with Distress in Elderly Cancer Patients: A Systematic Review

**DOI:** 10.3390/ijerph19063424

**Published:** 2022-03-14

**Authors:** Sandra Silva, Ana Bártolo, Isabel M. Santos, Anabela Pereira, Sara Monteiro

**Affiliations:** 1Center for Health Technology and Services Research (CINTESIS), Department of Education and Psychology, University of Aveiro, 3810-193 Aveiro, Portugal; isabel.santos@ua.pt (I.M.S.); smonteiro@ua.pt (S.M.); 2Center for Health Technology and Services Research (CINTESIS), Piaget Institute—ISEIT/Viseu, 3515-776 Viseu, Portugal; ana.bartolo@viseu.ipiaget.pt; 3William James Center for Research, Department of Education and Psychology, University of Aveiro, 3810-193 Aveiro, Portugal; 4Research Centre on Didactics and Technology in the Education of Trainers (CIDTFF), Department of Education and Psychology, University of Aveiro, 3810-193 Aveiro, Portugal; anabelapeireira@ua.pt

**Keywords:** cancer, elderly, depression, anxiety, distress

## Abstract

This study presents a systematic review of the sociodemographic, clinical, and psychosocial factors associated with distress in elderly cancer patients. Relevant studies were identified using four electronic databases: PubMed, Scopus, Web of Science and ProQuest. Cross-sectional and longitudinal studies exploring factors associated with distress in people over 60 years of age were included and independently assessed using the Joanna Briggs Institute Critical Assessment Checklists. A total of 20 studies met the inclusion criteria. Research showed that being a woman, being single, divorced or widowed, having low income, having an advanced diagnosis, having functional limitations, having comorbidities, and having little social support were factors consistently associated with emotional distress. Data further showed that the impact of age, cancer type, and cancer treatment on symptoms of anxiety and/or depression in elderly patients is not yet well established. The findings of this review suggest that the emotional distress of elderly cancer patients depends on a myriad of factors that are not exclusive, but coexisting determinants of health. Future research is still needed to better understand risk factors for distress in this patient population, providing the resources for healthcare providers to better meet their needs.

## 1. Introduction

Demographic aging has been a constant in recent decades worldwide. According to the United Nations, the number of elderly people aged 60 or over is expected to double by 2050 and more than triple by 2100, from 962 million in 2017 to 2.1 billion in 2050 and 3.1 billion in 2100 [1]. Considering these numbers, it is evident that the weight of the elderly in the population structure has increased significantly, due to the decrease in birth and mortality rates. While longevity is considered a positive phenomenon, it also means being more exposed to diseases, such as the cancer, which is currently one of the main causes of death worldwide [2]—10 million deaths in 2020 [3]. In 2040, the incidence of new cancer cases is expected to increase by approximately 64%, reaching 30.2 million new diagnoses and 16.3 million deaths per year. Of these new annual cases, about 50% will occur in people aged 65 and over, and more than 60% of cancer deaths will occur among individuals in this age category [4].

Therefore, it can be stated that aging is the main risk factor for the development of cancer. Despite this relationship, there was a shortage of clinical trials in people over 65 years of age. The same was observed with research in the social and behavioral areas focusing on young adults, leaving a gap in our understanding of the needs and impact of cancer on the quality of life and mental health of the elderly [5,6,7,8].

Depression and anxiety are the most common psychological disorders in people with cancer. Among the elderly population, the prevalence of depression is approximately 13.5% [9] and that of anxiety varies between 1.2% and 15% [10]; however, in the population diagnosed with cancer, these prevalences are slightly higher. The literature reveals a prevalence of depression of between 17% and 26%, varying according to the type of cancer, treatment phase and method of diagnosis [11,12]. For anxiety, the prevalence reported rates range from 2.5–23% [11]. However, the prevalence rates for these disorders in this population are estimated to be higher. This is because depression is generally undiagnosed and untreated, due to the overlap between the diagnostic criteria for depression and the symptoms generally attributed to cancer and the effects of treatment, such as sleep disturbances, decreased interest in genderual activity, and lack of energy [13]. In addition, the elderly have another peculiarity that makes it difficult to diagnose depression. They usually have somatic complaints (body pain and malaise) instead of affective complaints (sadness, guilt and self-criticism), in contrast to younger adults [14]. In turn, the diagnosis and treatment of anxiety has also been neglected in the context of cancer. Faced with a life-threatening illness (a stressful and traumatic event), anxiety can be considered an appropriate, expected and even normal reaction, and it can be difficult to determine when it is normative or pathological and should receive attention [15].

The presence of depression and/or anxiety has important repercussions in terms of therapeutic adherence and quality of life [16], which may influence the results of treatments [17], length of hospital stay [18], the hope of recovery [19] and mortality rates [20,21]. Therefore, it is important to understand the factors that increase the likelihood of the appearance of these disorders in elderly cancer patients. 

To the best of our knowledge, no study to date has systematically analyzed the factors associated with distress in the elderly population with cancer. In the current literature involving the general cancer population, demographic factors, such as age and gender, have been associated with an increased risk of depression. Furthermore, a variety of cancer-related factors have been shown to affect the development of depression and anxiety, including the type of cancer, type of treatment, and stage. Social and economic factors, such as unemployment, low education, and lack of social support, have been associated with high levels of depression. Finally, several psychological factors, such as a history of mental illness and personality factors, namely neuroticism, were also associated with distress [22]. Based on these findings, this study aimed to conduct a narrower review of the existing literature on the relationship between sociodemographic, clinical, and psychosocial factors and emotional distress of elderly people with cancer.

## 2. Materials and Methods

This review was conducted according to the Preferred Reporting Items for Systematic Reviews and Meta-analysis (PRISMA) 2020 guidelines [23] and was registered in the international prospective register of systematic reviews (PROSPERO) and is available in full on the NIHRHTA program website (https://www.crd.york.ac.uk/prospero/display_record.php?ID=CRD42021242343, accessed on 16 April 2021).

### 2.1. Eligibility Criteria

Studies were included in the systematic overview if they: (i) involved cancer patients aged 60 years or older; (ii) included patients at the active stage or disease-free survivors; (iii) identified sociodemographic, clinical, and psychosocial characteristics associated with psychological distress (emotional suffering, with symptoms of depression and anxiety); (iv) were written in English; and (v) were published in a peer-reviewed journal between January 2015 and March 2021. Literature/systematic reviews, validation studies, book chapters, unpublished articles, commentaries, and conference abstracts were excluded.

### 2.2. Literature Search

A systematic search was performed using Scopus, Web of Science, PubMed, and ProQuest. The following key terms were used: cancer/oncology, older/elderly/geriatric, distress/depression/anxiety, relationship. The search was adapted for each of the databases and the OR and AND functions were used, as well as field labels (title, abstract and text). Specific filters related to publication date, language, and document type were used whenever possible. Searches in these databases were supplemented by a manual search of the reference lists of included articles. The first search was performed in March 2021 and was then rerun in December 2021 to identify possible further studies. To obtain the unavailable articles, the relevant authors were contacted.

### 2.3. Extraction and Synthesis Strategy

The selection process of the articles was conducted by the first author, taking into consideration the eligibility criteria defined by the team. The author performed an exhaustive review of all titles and abstracts obtaining a list of articles for reading the full text. All documents raising any doubts were discussed and resolved by consensus between all the co-authors.

### 2.4. Quality Appraisal

The quality of retrieved articles was critically assessed using the Joanna Briggs Institute (JBI) Critical Review Checklists for cross-sectional analytical studies and cohort studies [24,25]. Each item on these checklists was appraised as “yes”, “no”, “unclear” or “not applicable”. A substantial number of the checklist criteria had to be met to include each study in the review, i.e., at least 50% of the JBI criteria. Any disagreements between the reviews were resolved by discussion among all co-authors.

## 3. Results

### 3.1. Search Results

A flowchart of the literature search is shown in Figure 1. A total of 770 potentially relevant articles were identified (761 studies were identified via databases and registries, and 9 via other methods). From these, 132 duplicate articles were removed. In the next phase, titles and abstracts were analyzed, resulting in the exclusion of 587 articles. Most of the studies were excluded because they did not assess the association between sociodemographic, clinical, and psychological factors and distress in elderly people with cancer. Therefore, only 51 full texts remained for eligibility verification. Of these, 21 articles had a sample consisting of individuals under 60 years of age, and 9 articles did not assess associations between factors or assessed other associations that were not of interest to this review. One of the articles evaluated associations between the intended factors, but with a sample composed of individuals with different morbidities and not just cancer. Finally, another article was excluded, as the sample consisted of individuals with different mental disorders and not just anxiety and/or depression. No studies were excluded based on the critical assessment tools.

### 3.2. Study Characteristics

Twenty studies were included in the literature review, of which 15 were cross-sectional and five were cohort studies.

A large number of the studies were carried out in the United States (*n* = 8); the remaining studies were carried out as follows: France (*n* = 3), Israel (*n* = 2), Germany (*n* = 1), Iran (*n* = 1), Japan (*n* = 1), Jordan (*n* = 1), Norway (*n* = 1), China (*n* = 1) and Turkey (*n* = 1). 

The sample size of the different studies ranged between 42 [26] and 53,821 [27] (M = 3904; MD = 252.5; SD = 12,079) (see Table 1). This wide range in sample sizes among the different studies is due to the way in which the samples were obtained. In some studies conducted in the United States, the sample size was larger because researchers collected participant data through an electronic database, i.e., SEER (Surveillance, Epidemiology and End Results)—Medicare. 

Concerning participant characteristics, and gender in particular, it was found that the average frequency of male participants was 53.5% and 48% for female. The definition of elderly was not consensual; in some studies, the term elderly was applied to someone over 60 years of age while in others, it was someone over 65 or 70 years of age. The age of participants included in the studies ranged between 60 and 98 years (M = 70.8; SD = 5.6). A single study referred to the mean age of participants (61.9 years) at the time of diagnosis of the oncological disease [38].

Level of education was mentioned in 13 of the studies. On average, 5.1% of individuals were illiterate and 27.4% had higher education.

From the studies reporting marital status (17/20), it was found that 66.3% of the subjects in the sample were married or living with a partner, while 32.1% were single, widowed or divorced.

The type of cancer present in each sample was highly variable, with some studies selecting people with a specific diagnosis: hematological [28], gynecologic [30], non-Hodgkin´s lymphoma [26], colorectal [41], gastrointinal [45], and bladder [46]; and others including any type of cancer [27,29,32,33,34,37,38,39,40,42,43,44]. Despite this variability, breast/gynecological, gastrointestinal, and colorectal cancers were the most represented, with 20.9%, 17.7% and 16.4%, respectively. 

Regarding the stage of the disease at the time of diagnosis, 14 studies reported this, showing that 62.9% of individuals were diagnosed between stage 0 and III of the disease and that only 34.3% were diagnosed at a more advanced stage (stage IV), characterized by the presence of metastases. 

Another variability present in the samples from different studies concerned the phase of the oncological disease in which participants were at the time of data collection. In most studies, participants were in the active phase of the disease and/or in treatment [26,27,29,32,33,34,35,39,40,43,44,45]. However, some studies included individuals who were diagnosed up to 5 years prior [28], while others included individuals diagnosed 5 years or more prior [36,38]. Some studies did not discriminate which phase of the disease the participant was in at the time of data collection [30,33,37,46].

### 3.3. Measures

The measures used to assess anxiety and depression were very heterogeneous. Regarding the measures used for depression, six studies used the geriatric depression scale (GDS) or a reduced version of it [32,34,37,43,44,45], three used the Veterans RAND 12-item Health Survey (VR-12) [30,41,46], three used the Hospital Anxiety and Depression Scale (HADS) [29,35,39] and two used the Center for Epidemiological Studies Depression Scale (CES-D) [36,38]. The Patient Health Questionnaire-9 (PHQ-9) [28], Mini International Neuropsychiatric Interview (MINI) [26] and Edmonton Symptom Assessment System Revised (ESAS-r) [42] were three other measures used to assess depression. In another study, the diagnosis of depression was identified through the codes of the International Statistical Classification of Diseases—Ninth Edition presents in the SEER database [27]. Anxiety was assessed in three studies by HADS [29,35,39] and in others by General Anxiety Disorder-7 (GAD-7) [28], Profile Of Mood States (POMS) [38] and ESAS-r [42]. Distress in three studies was measured using an 11-point distress thermometer [32,33,44].

### 3.4. Factors Affecting Psychological Distress

The factors associated with distress that were found in the various articles were quite heterogeneous. Thus, we tried to group these factors into three distinct groups: sociodemographic, clinical, and psychosocial. Within the clinical category were factors related to cancer and treatment and factors related to health in general.

#### 3.4.1. Sociodemographic Factors and Distress

Nineteen of the 20 studies included investigated the association between sociodemographic factors and emotional distress. Of these, eight did not find any association between the variables. Age, gender, marital status, education, income, and race were the main sociodemographic factors included in the analyses.

Associations between age and distress were examined in 16 studies, but only four significant results were found. Three studies showed results in the same direction, that is, older age was a predictor of lower scores for depression [30,46] and anxiety [29]. However, Goldzweig et al. [32] found that depression among the oldest group was 2.8 times higher than among the younger-old group, and distress among the oldest group was 1.95 times higher in comparison to distress among the younger-old.

Gender was a sociodemographic factor included in the analyses of 14 studies, but only two revealed a statistically significant association. Alwhaibi et al. [27] observed that women diagnosed with colorectal cancer had a 46% higher risk of newly diagnosed depression compared to men with the same type of cancer. Similar results were found in the study by Solvik et al. [42], in which women reported significantly higher scores of depression and anxiety.

Marital status was evaluated in eight studies. The results found were relatively consensual. Hong et al. [33] found that married individuals suffered lower levels of distress, and in the same way, Meier et al. [28] observed that the lack of a partner (single, divorced, and widowed) was associated with higher levels of depression. Oserowsky et al. [46] also found that being married was inversely associated with a positive depression screening. The results presented by Ladaninejad et al. [37] partially corroborated those mentioned above, that is, higher depression scores were observed among widowed patients, while lower scores were more common among single patients.

Education was a factor examined in 50% of the selected articles, but only two studies reported significant results. Hong et al. [33] found that patients with a degree of junior high school or lower exhibited the highest level of distress, followed by high school and technical secondary school graduates. The patients with junior college and higher education scored the lowest regarding level of distress. In line with these results, Oserowsky et al. [46] showed that a higher education level was associated with a positive depression screening. 

Although less analyzed, four studies included race, but only two showed statistically significant results. Survivors of nonwhite colorectal cancer were 51% more likely to develop depressive symptoms [41]; these results were supported by Oserowsky et al. [46], who found that nonwhite race was an independent predictor of a positive depression screening.

Finally, income was analyzed in five studies, of which three revealed significant results. The lowest depression levels were observed in elderly patients whose incomes matched their expenses, while the highest were observed among those whose incomes were less than their living expenses [37]. Clark et al. [41] and Oserowsky et al. [46] found that the same results, i.e., income less than US$300,000 per year, were predictors of positive depression screening [41,46].

#### 3.4.2. Clinical Factors and Distress

Clinical factors can be classified into two groups: those that are directly related to cancer and those that, although they may be a consequence of the oncological disease and treatment, may also be present in the geriatric population in general. The most analyzed clinical factors in different studies that are directly related to cancer were type of cancer, stage of cancer, metastasis, type of treatment, time between diagnosis and research.

The association between the type of cancer and distress was evaluated in 12 studies, although significant results were found in only two, and these were not congruent. In the study by Alwhaibi et al. [27], colorectal cancer was associated with higher levels of depression. The highest percentages of new depression were among women with colorectal cancer compared to those with breast cancer, and among men with colorectal cancer compared to those with prostate cancer. In opposition to these results, Ladaninejad et al. [37] found that colon cancer patients had significantly lower depression scores than those with other types of cancer (esophageal, breast, prostate, lung, stomach, head, and neck). 

The medical system for classifying tumors, depending on their extension and spread throughout the body, was not the same in all studies. However, the results found were similar. The risk of diagnosed depression was higher among elderly cancer patients diagnosed at an advanced stage compared with those diagnosed at an early stage. Elderly individuals diagnosed with cancer at stage IV had a 63% higher risk of being diagnosed with depression compared with those diagnosed at stage I [27]. In the study by Klapheke et al. [30], later stage at diagnosis was significantly associated with greater odds of depression, and in the study of Wiesel et al. [29], advanced stage of the disease was shown to be a significant predictor of depression but not anxiety. 

In two of the 12 studies that investigated an association between treatment and distress, significant results were found. Elderly people with hematologic cancer who were undergoing chemotherapy at the time of data collection had significantly higher levels of depression compared to those who had previously received chemotherapy, radiotherapy, or transplantation [28]. However, when chemotherapy was effective, the risk of depression was lower [34]. In addition to the type of treatment, its duration was also investigated by Malak et al. [35], who concluded that a shorter duration of treatment predicted higher scores in terms of depressive symptoms. 

The presence of metastases was evaluated in two studies and the time elapsed between diagnosis and data collection was evaluated in seven. However, no significant associations were found. 

As for clinical variables related to health in general, limited/deficient mobility, limited activity, functional difficulties, deficiencies in Activities of Daily Living (ADL) and poor performance were different but closely related terms used by the authors [26,28,30,32,34,36,37,38,40,41,42,43,46]. Limited mobility and the need for care were associated in hematological cancer patients with depression [28]. Identical results were found by Canoi-Poitrine in a sample consisting of different types of cancer patients [40]. Baeza-Velaso et al. [26], applying the scale of the Eastern Cooperative Oncology Group (ECOG), found differences which were statistically related to performance status among patients with major depressive disorder. The percentage of participants with a high reduction in activities and more time in bed was higher than that of participants with a slight reduction in activities (85.7% vs. 14.3%, *p* < 0.05). Regarding the three studies that assessed difficulties in ADLs, the results found by Klapheke et al. [30] in elderly women with gynecological cancer, by Clark et al. [41] in elderly with colorectal cancer, and by Oserowsky et al. [46] in elderly people with bladder cancer went in the same direction; that is, for every additional impairment in ADL, the odds of depressive symptoms increased. Lastly, Deimling et al. [38] claimed that functional difficulties were symptoms not attributable to cancer, but rather, were strongly related to anxiety and depression.

Eight studies reported an association between the presence of comorbidities and distress. Clark et al. found that patients with multiple comorbidities were more likely to have a positive depression screening. However, the results of Meier et al. [28], Wiesel et al. [29], and Deimling et al. [38], in addition to the association between the presence of comorbidities and depression, also identified an association between the presence of comorbidities and anxiety. Other authors investigated the cancer comorbidity spectrum and found that cardiovascular diseases, sciatica [30], stroke [30,46] diabetes, respiratory disease [37], muscular disease, and urinary issues [46] had a significant relationship with depression. Soto-Perez-de-Celis et al. [39] looked at another type of disease/dysfunctionality which, according to the authors, was also often present in the elderly and which ended up being comorbid with cancer. In this study, one third of the elderly diagnosed with cancer reported at least one sensory impairment. Patients with hearing impairment, visual impairment, and dual sensory impairment were associated with depression. Hearing impairment and dual sensory impairment were significantly associated with anxiety. 

Pain is a clinical factor evaluated differently by different authors, but the results were consensual. Higher scores for pain related to cancer and/or other comorbid conditions were associated with higher levels of depression [40,43] and anxiety [42]. In addition to cancer-related pain, Canoui-Poitrine et al. [40] also found that polypharmacy (≥5 nonanti-depressant drugs per day) was associated with clinical depression.

Finally, malnutrition was a factor evaluated in four of the included studies, but only two reported significant results. Meier et al. [28] found poor nutrition to be associated with higher levels of anxiety, and Duc et al. [34] found malnutrition to be associated with depressive symptoms.

#### 3.4.3. Psychosocial Factors and Distress

In the selected literature, few studies investigated the relationship of psychosocial factors with distress. Among the main factors examined were social support, hope, emotional and cognitive function, personality characteristics, and a history of depression.

Although named differently by different authors, the concept of social support or social functioning was evaluated in eight studies and four reported significant results. The measures used by the authors to assess social support were all different. Wiesel et al. [29] used the Medical Outcomes Study (MOS), a measure of perceived availability of social support, that showed that depression and anxiety were significantly predicted by inadequate social support. Canoui-Poitrine et al. [40] defined inadequate social support as the lack of a primary caregiver or support from family and friends capable of meeting the patient’s needs. They found identical results, that is, inadequate social support was independently associated with depression. Meier et al. [28] used the disease-specific social support scale (SSUK-8), which assesses positive support versus detrimental interaction, the Luben social network scale (LSNS-6), which assesses social isolation, and EORTC Quality of Life Questionnaire (EORTC QLQ-C30), which includes a social functioning subscale. Concerning anxiety, they did not find an association between anxiety and lack of social integration, but found an association between anxiety and detrimental social interaction. A strong negative association with the presence of depressive symptoms was found for the QoL function scales including social function. Okumura et al. [45] evaluated preoperative social frailty in elderly patients with gastrointestinal cancer. One year after surgery, they applied five dichotomic questions and found that preoperative social frailty was associated with new depressive symptoms one year after surgery. 

As previously mentioned, Meier et al. [28] used the EORTC QLQ-C30, which includes several subscales, one of which is the assessment of cognitive function. A strong negative association was found between the cognitive function subscale and the presence of depressive symptoms. The Mini Mental State was the scale chosen by Canoui-Poitrine et al. [40] to assess cognitive status. Cognitive impairment was a factor independently associated with clinical depression. In the study by Ladaninejad et al. [37], cognitive status was assessed by the Mental Test Score (MTS), while in the studies by Baeza-Velasco et al. [26] and Duc et al. [34], it was assessed by the Mini Mental State (MMS). These studies did not reveal an association with distress.

Hope was a factor included in the analyses by Goldzweig et al. [44] and Malak et al. [35]. It was assessed by the Adult Hope Scale and Herth Hope Index, respectively. Goldzweig et al. [44] found that patients’ hope of being cured was a significant predictor of levels of distress (negative correlation). Similar results by Malak et al. [35] revealed that hope was a significant predictor of depression but not anxiety.

In a sample of breast, prostate, and colorectal cancer survivors (long-term 5+ years), three personality dimensions were statistically significant predictors of depression. Neuroticism had the strongest independent effect, followed by conscientiousness and agreeableness.

Solvik et al. [42] found a strong correlation between anxiety and depression (0.76), and in the study by Malak et al. [35], anxiety emerged as one of the main predictors of depressive symptoms. Finally, a history of major depressive disorder was significantly more frequent among participants with current Major Depressive Disorder (MDD) than those without; almost 70% of patients with current MDD already had depression in the past [26]. Depressive symptoms at baseline (before the first cycle of chemotherapy) were associated with a higher risk of depression after completion of treatment (four cycles of chemotherapy) [34].

### 3.5. Study Quality Assessment

Descriptions of the critical appraisal are shown in Table 2 and Table 3. Most cross-sectional studies met the JBI criteria. The most common reasons for bias in the results of the qualitative evidence assessment were related to the lack of control for confounding factors in approximately 50% of the studies. The self-report measures used in the included reports proved to be valid tools in previous research; however, only 30% reported the psychometric characteristics of the instruments. Regarding longitudinal studies, most also met the JBI criteria, except that not all participants were free of the outcome (depressive symptoms) at baseline.

## 4. Discussion

The present study aimed to review the association between emotional distress (symptoms of depression and anxiety) and sociodemographic, clinical, and psychosocial aspects in elderly people diagnosed with cancer. The study involved men and women over 60 years of age diagnosed with different cancer types. The most prevalent were breast/gynecological cancer, colorectal cancer and gastrointestinal cancer.

The relationship between sociodemographic and clinical factors and distress was analyzed in all selected studies, except in the study of Soto-Perez-de-Celis et al. [39]. The relationship between distress and psychosocial factors was analyzed in 14 studies [26,28,29,30,32,34,35,36,37,40,41,43,44,45].

Regarding sociodemographic factors, studies have consistently demonstrated significant associations between gender, education, and income, and depressive symptoms, suggesting that being a woman, having a low education and having low incomes are risk factors for depression. These findings are not surprising and are in line with results previously found by Maier et al. [47] in a literature review that included studies involving elderly people (≥65 years), but where the diagnosis of cancer was not an inclusion criterion. This may indicate that these variables can affect the emotional state of elderly people in general, regardless of the diagnosis of cancer. Still in that review, the white race was considered a protective factor for depression. In our review of the four studies that explored this effect, two revealed congruent results. Concerning age, the results found in three out of four studies are similar, that is, older age is considered a protective factor for anxiety, depression and distress. However, Goldzweig et al. [32] found precisely the opposite in relation to depression and distress, making it clear that the influence of age as a predictive factor for depression and distress was more pronounced at 86 years of age. In the literature review by Maier et al. [47], the 5 out of 16 studies carrying significant results have showed that age was associated with a greater risk of depression, and these results were congruent with those of Goldzeig et al. [32]. However, given this disparity of results present in both reviews and knowing that the review by Maier et al. [47] does not focus on people diagnosed with oncological disease, it is not possible to draw conclusions regarding the age factor. 

Regarding clinical variables not directly related to the diagnosis of cancer, seven of the 13 studies included showed a significant relationship between limited mobility, performance status, functional difficulties, impairment in ADLs and depression and/or anxiety. The findings of this review are in agreement with the review by Maier et al. [47], except with regard to impairment in ADLs, that Maier only identified them as a risk factor in one of six studies, suggesting less influence of this factor on depression.

The presence of multiple comorbidities is revealed in this review as a risk factor for depression and anxiety. We found that cardiovascular disease, stroke, sciatica, diabetes, respiratory disease, muscular disease, and urinary issues have a significant relationship with depression. However, in the review by Maier et al. [47], there was no congruence in relation to stroke and heart disease. Studies demonstrated that these pathologies are a risk factor for depression, others have not found any significant association, however, it is important to mention that these results may depend on the number of months/years elapsed from the onset of the disease until the investigation. Considering the objective of our review, it is important to emphasize that in Maier’s review, newly diagnosed cancer and ongoing cancer were not significant risk factors associated with distress. In our review, higher scores for pain related to cancer and/or other comorbid conditions were associated with higher levels of depression and anxiety. These results are supported by different literature reviews. Massie et al. [48] found that the prevalence of anxiety and depression was significantly higher in cancer patients with pain than for those without pain. Parpa et al. [49] found that higher levels of anxiety have also been related to pain, or even with greater use of pain-relieving medications, however, anxiety disorders associated with pain in elderly cancer patients decreased after treatment with analgesics. Severe and emergent pain were identified as risk factors, although chronic pain does not reveal significant results [47]. Finally, although no significant results were found in all studies in which the nutrition factor was included, the results point to a higher risk of depression and anxiety in people with poor nutrition, however, in the aforementioned reviews, none of the studies included this variable in the analysis so these results could not be compared.

The measures used by the authors to assess social support were all quite different and, furthermore, the fact that they are self-response measures makes the results more subjective and difficult to interpret. However, depression was associated with perceived inadequate social support [29,40,45] and anxiety was associated with perceived inadequate social support [29] and detrimental social interaction [28]. The review by Maier et al. [47] points in the sense that higher scores on social network measurement scales are associated with a lower risk of depression, although “loneliness” and “negative family interaction” were assessed in more than one study and did not produce consensual results. 

In two of the five studies analyzed, significant but not congruent associations were found between cognitive status and depression. Regarding this factor, once again the results presented by Maier et al. corroborate ours [47]. Patient’s hope being cured is a significant predictor of patient lower levels of distress and depression [35,44], but not of anxiety [35]. The reviews that have been mentioned do not include studies that assess the association between distress and hope, so these results could not be compared. The same happened in relation to personality type, there is no literature to compare the association we found between neuroticism and depression.

A history of depression before the diagnosis of the oncologic disease was evaluated in only one study, verifying a positive association [26]. In another study, depressive symptoms were assessed before the start of the first chemotherapy cycle and after the end of the four chemotherapy cycles, and an association between depressive symptoms before chemotherapy and depression at the end of treatment was also found [34]. In relation to this variable, as well as in relation to the history of anxiety, the results found by Maier et al. [47] were not consensual. In our review, no shield included the history of anxiety in the analysis, however anxiety at the time of data collection appeared to be strongly correlated with depression [42] and emerged as one of the main predictors of depressive symptoms [28].

It is possible that greater homogeneity in the samples included in the different studies and greater homogeneity in the instruments for measuring independent and dependent variables would lead to more consensual and comparable results.

### Limitations and Future Directions

Some limitations of this review should be acknowledged. This review only includes articles published in English, which may have led to the exclusion of relevant studies in the research. Given the scarcity of existing literature in the age group above 65 years, the inclusion criteria for this study became very comprehensive, which limited some comparisons. Studies were included with samples consisting of individuals with any type of cancer, at any stage of the disease, stage of treatment and stage of disease, from newly diagnosed to long-term survivors (>10 years). Another weakness of this review concerns the measures used to assess depression, anxiety and distress, which were quite diversified and almost always self-reported, which provides less reliable results than structured interviews. Finally, a large part of the studies included in this review presented a cross-sectional design, which does not allow making inferences of causality. Future studies will benefit from longitudinal designs, particularly to explore the trajectories of emotional distress in elderly cancer patients.

## 5. Conclusions

Given the high prevalence of oncological disease in individuals over 65 years of age, and knowing that depression and anxiety are the most common psychological disorders in this population and that they affect not only treatment adherence but also quality of life, it is essential to identify risk and protective factors associated with developmental disorders. Therefore, this review provides a reference to assist researchers and healthcare providers in supporting elderly cancer patients, and facilitates the establishment of referral paths for patients at higher risk of distress. Furthermore, the findings of this systematic synthesis identify research gaps that need further exploration, namely, the impact of clinical factors related to cancer on the distress levels of this patient population.

## Figures and Tables

**Figure 1 ijerph-19-03424-f001:**
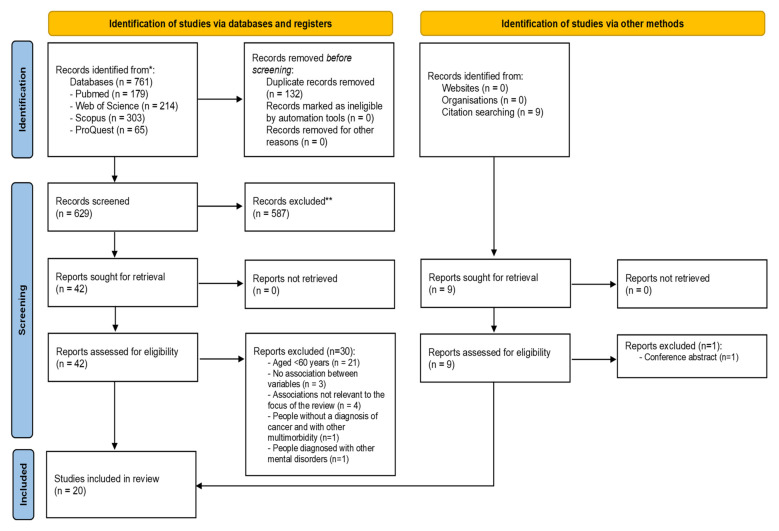
PRISMA flow diagram of the selection process.

**Table 1 ijerph-19-03424-t001:** Descriptive characteristics and results of the included studies.

Ref ID	Author (Year)	Country	Study Design	Sample Size (N)	Cancer Type	Mean Age	% Female	Distress Measures	Main Factors	Main Results
[28]	Meier et al. (2020)	Germany	Cross-sectional study	N = 425	Hematological	75.7 (4.2)	38.4%	General Anxiety Disorder (GAD-7) Patient Health Questionnaire (PHQ-9)	Gender, age, partnership, education, type of cancer, disease condition, treatment, malnutrition, polypharmacy, tendency to fall, limited mobility, care level, self-perceived social support, social isolation, quality of life (QoL).	- ↓ mobility, need for care, comorbidity, ongoing chemotherapy, lack of partnership and ↓QoL were associated with ↑ depression. ↓ social interaction, cognitive and emotional functioning, poor nutrition, and comorbidity was associated with ↑ anxiety.
[27]	Alwhaibi et al. (2017)	USA	Retrospective Cohort Study	N = 53,821	Breast, colorectal, prostate	Age groups (66–69 y, *n* = 14,007), (70–74 y, *n* = 15,791), (75–79 y, *n* = 11,276), (≥80 y, *n* = 2747)	48.9%	International Classification of Diseases, Ninth Revision, Clinical Modification (ICD-9-CM)	Cancer type, stage at cancer diagnosis, treatment, gender.	- Women with colorectal cancer (CRC) compared with men with CRC; women with CRC compared with those with breast cancer, and among men with CRC compared with those with prostate cancer; and survivors diagnosed at an advanced stage were associated ↑ % diagnosis depression.
[29]	Wiesel et al. (2015)	USA	Cross sectional- study	N = 500	Lung, gastrointestinal, gynaecological, breast, genitourinary and others.	73.1 (6.18)	56.2%	Hospital Anxiety and Depression Scale (HADS)	Age, gender, education, cancer type, stage of disease, comorbidities, social support.	- ↑ age, ↑ social support, ↓ number of comorbidities were associated ↓ anxiety. - ↓ social support, ↑ number of comorbidities, and advanced stage were associated with ↑ depression.
[26]	Baeza-Velasc et al. (2017)	France	Cross sectional-study	N = 42	Non-Hodgkin’s lymphoma	81.6 (4.2)	52.4%	Mini-international neuropsychiatric interview (MINI)	Gender, age, marital status, number of children, schooling, geographical area, stage, comorbidities, No of drugs, malnutrition, performance status, fatigue, history of depression, cognitive status coping strategies, perception of social support, conflict, depth, global health status and global QoL.	- ↓ self-perceived ↓ performance status, ↑ fatigue and history of depression were associated a patients with Major of Depressive Disorder (MDD).
[30]	Klapheke et al. (2019)	USA	Cross-sectional study	N = 11,862	Gynecologic	age all cancer = 74.8 (6.5), age no cancer = 75.3 (7.0).	100%	Algorithm by Rost et al. [31] and responses to questions from the Diagnostic Interview Schedule in the MHOS. - Veterans RAND 12-item Health Survey (VR-12)	Age, race, education level, marital status, income, region, cancer site, comorbidities, stage at diagnosis, Activitie Daily Living (ADLs), time since diagnosis, health-related quality of life.	- ↑ age was associated with ↓ depressive symptoms. - later stage diagnosis, cardiovascular disease, stroke, sciatica, impairment in ADL, ↓ physical and mental measures of HRQOL were associated ↑ depressive symptoms.
[32]	Goldzweig et al. (2018)	Israel	Cross-sectional study	N = 243	Lung, prostate, breast, colorectal, melanoma, other	77.53 (9.29)	35.8%	Geriatric Depression Scale (GDS-5) Distress thermometer (1 item)	Age, gender, time of diagnosis, stage of cancer, performance status, comorbidity treatment, social support.	- ↑ age was associated ↑ depression levels and distress.
[33]	Hong et al. (2015)	China	Cross-sectional study	N = 153	Digestive, respiratory, breast, urogenital system, others	67.2 (6.01)	39.2%	Distress thermometer	Gender, marital status, education, income, disease site of cancer, treatment.	- married, ↑ education, ↑ monthly income had ↓ distress.
[34]	Duc et al. (2017)	France	Prospective cohort study	N = 260	Colon, stomach, pancreas, non-Hodgkin´s lymphoma, prostate, ovary, bladder, lung, unknown primary origin	77.6 (4.8)	44.6%	GDS-15	Age, gender, live alone, education, marital status, cancer site, performance, advanced disease, treatment, ADLs, Instrumental Activities of Daily Living (IADLs), cognitive status, nutritional status, fall risk.	- Depressive symptoms at baseline, and malnutrition was associated ↑ risk of depression. - Effective chemotherapy was associated with a ↓ risk of depression.
[35]	Malak et al. (2020)	Jordan	Cross-sectional study	N = 150	Undefined	64.33 (3.46)	42%	HADS	Age, educational level, duration of cancer treatment, type of treatment, health insurance, hope, anxiety.	- ↓ Duration of treatment, ↓ hope, and ↑ anxiety were the predictors of ↑ depression.
[36]	Deimling, et al. (2017)	USA	Cross sectional study	N = 275	Breast, prostate	73.18 (7.18)	58.2%	Center for Epidemiologic Studies Depression Scale (CES-D)	Age, gender, race, personality, type, stage at diagnosis, years since diagnosis, No of treatments, No of symptoms, No of symptoms attributed to cancer, No of health conditions, functional difficulties, cancer-related health insurance.	- Neuroticism, conscientiousness, agreeableness were significant predictors of depression.
[37]	Ladaninejad et al. (2019)	Iran	Cross-sectional study	N = 200	Colon, esophageal, breast, prostate, lung; head and neck, gastric	67.82 (6.73)	51%	GDS	Gender, marital status, living with, frequency of contact with children, education, income, type of underlying disease, type of cancer, stage of cancer, metastasis, pain, nausea, vomiting, shortness of breath, hair loss, frequency of chemotherapy, ADLs, cognitive status, perceived social support.	- Single patients and patients with colon cancer had ↓ depression. - Widowed, elderly, ↓ income, with diabetes and respiratory diseases had a ↑ depression.
[38]	Deimling et al. (2017)	USA	Cross sectional design	N = 245	Breast, prostate, colorectal	75.9	63%	CES-D Profile of Mood States (POMS)	Age, type of cancer, years since diagnosis, comorbidities, functional difficulties, current cancer and non cancer symptoms, worry dimensions, psychological distress.	Symptoms not attributed to cancer, functional difficulties, No of comorbidities are relatively strong correlates of depression and anxiety.
[39]	Soto-Perez-de-Celis et al. (2015)	USA	Prospective study	N = 750	Lung, gastrointestinal, breast, gynecological and others	72 (median)	44%	HADS	Sensory impairments	Sensory impairments were associated with depression and anxiety.
[40]	Canoui-Poitrine et al. (2015)	France	Cross-sectional study	N = 1092	Ovarian and endometrial, esophagus, prostate, urinary, colorectal, breast, skin, unknown primary, hematological, stomach, lung, pancreas, and others	80.4 (5.7)	48.8%	Semi-structured interview was designed to identify eight of nine symptoms of diagnostic and Statistical Manual of Mental Disorders—IV (DSM-IV) criteria for a major depressive	Age, gender, living alone, with inpatient status, metastasis, mobility, functional status, pain, malnutrition, cognitive impairment, comorbidities, No of nonantidepressant drugs, polypharmacy, social support.	Inpatient status, inadequate social support, impaired mobility, cognitive impairment, polypharmacy, and cancer-related pain were associated depression.
[41]	Clark et al. (2016)	USA	Retrospective cohort study	N = 1785	Colorectal	78 (7)	51%	Depression was defined as an affirmative answer to at least one of the three depression screening questions;—VR-12.	Age, race, gender, education, income, homeownership, marital status, tumor size, stage, and radiation therapy, No of months from CRC diagnosis to survey, No of comorbidities, impairment ADLs, age per 10 years)	Nonwhite race, ↓ income, comorbidities, impairment in ADLs were associated with depression.
[42]	Solvik et al. (2020)	Norway	Cross-sectional study	N = 174	Breast, prostate, lymphoma, lung, colon, brain, rectal, bladder, ovarian and others	77.4 (7.1)	41%	Edmonton Symptom Assessment System Revised (ESAS-r)	Age, civil status, education, type of cancer, time since diagnosis, ongoing treatment, functional level, body mass index, fatigue, anxiety.	- ↑ pain was associated with higher scores of fatigues and anxiety and the women reported higher levels the anxiety and depression. - strong correlation between anxiety and depression.
[43]	Atag et al. (2018)	Turkey	Prospective study	N = 170	Lung, gastrointestinal, breast, gynaecologic, genitourinary and other	71.19 (5.03)	47.1%	GDS	Age, gender, marital status); awareness of disease, stage, No of comorbidities, pain, time since diagnosis operated due to cancer, radiotherapy, social support.	- ↑ pain in patients with depressive symptoms.
[44]	Goldzweig et al. (2017)	Israel	Cross-sectional design	N = 90	Prostate, lung, colorectal, breast	Patients 90.49 (2.40); Spouses 84.96 (9.87)	Patients = 55.6%; Spouses = 44.4%;	GDS Distress thermometer—1 item	Age of the patient, age of the caregiver, comorbidity, treatment, social support, hope	- ↑ patient´s age and ↓ the patient´s hope being cured were predictors of distress.
[45]	Okumura et al. (2020)	Japan	Cohort study	N = 48	Gastrointestinal	71	33%	GDS	Age, gender, marital history, level of education, depression at baseline, clinical stage, cancer type, performance status, complication, postoperative, adjuvant therapy, social frailty.	- Preoperative social frailty was associated with new-onset depressive symptoms.
[46]	Oserowskyet al. (2021)	USA	Retrospective cohort study	N = 5787	Bladder	77.4 (6.8)	24%	Affirmative answer to at least one of the three depression screening questions;—VR-12.	Age, race, gender, education, income, marital status, smoking status, and homeownership, cancer stage, ADLs, self-reported comorbidities, general health.	- ↑Age, married, higher education were associated with a ↓ depression. General health, nonwhite race, income <$30,000, difficulties with ADL, stroke, muscular disease, and urinary issues were predictors of depression.

**Table 2 ijerph-19-03424-t002:** Critical appraisal of the included studies—Analytical cross-sectional studies.

Criteria/Studies	[43]	[26]	[40]	[36]	[38]	[44]	[32]	[33]	[30]	[37]	[35]	[28]	[42]	[39]	[29]
Inclusion criteria clearly defined	Yes	Yes	Yes	Yes	Yes	Yes	Yes	Yes	Yes	Yes	Yes	Yes	Yes	Yes	Yes
Detailed description of subjects and setting	Yes	Yes	Yes	Yes	Yes	Yes	Yes	Yes	Yes	Yes	Yes	Yes	Yes	Yes	Yes
Exposure measured in a valid and reliable way	Yes	Yes	Yes	Yes	Yes	Yes	Yes	Yes	No	Yes	Yes	Yes	Yes	Yes	Yes
Objective criteria for measurement of the condition	Yes	Yes	Yes	Yes	Yes	Yes	Yes	Yes	Yes	Yes	Yes	Yes	Yes	Yes	Yes
Confounding factors identified	No	No	Yes	Yes	No	No	Yes	No	Yes	Yes	No	Yes	No	Yes	Yes
Strategies for dealing with confounders	No	No	Yes	Yes	No	No	Yes	No	Yes	No	No	Yes	No	Yes	Yes
Results measured in a valid and reliable way	Yes	Yes	Yes	Yes	Yes	Yes	Yes	Yes	No	Yes	Yes	Yes	Yes	Yes	Yes
Appropriate statistical analysis	Yes	Yes	Yes	Yes	Yes	Yes	Yes	Yes	Yes	Yes	Yes	Yes	Yes	Yes	Yes

**Table 3 ijerph-19-03424-t003:** Critical appraisal of the included studies—cohort studies.

Cohort Studies	[45]	[27]	[34]	[41]	[46]
Two groups similar and recruited from the same population	Yes	Yes	Yes	Yes	Yes
Exposures measured similarly to assign people to both exposed and unexposed groups	Yes	Yes	Yes	Yes	Yes
Exposure measured in a valid and reliable way	Yes	Yes	Yes	Yes	Yes
Objective criteria for measurement of the condition	Yes	Yes	Yes	Yes	Yes
Confounding factors identified	Yes	Yes	Yes	Yes	Yes
Strategies for dealing with confounders	Unclear	Yes	Yes	Yes	Yes
Participants free of the outcome at the start of the study	Yes	Yes	No	Unclear	Unclear
Outcomes measured in a valid and reliable way	Yes	Yes	Yes	Yes	Yes
The follow up time reported and sufficient to be long enough for outcomes to occur	Yes	Yes	Yes	Yes	Yes
Complete follow-up or presentation of the reasons for the loss of follow-up	Yes	Yes	Yes	Unclear	Unclear
Strategies to address incomplete follow up utilized	Yes	Yes	Yes	Unclear	Unclear
Appropriate statistical analysis	Yes	Yes	Yes	Yes	Yes

## Data Availability

Not applicable.

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
