# Peer review of "Towards a Better Understanding of the Factors Associated with Distress in Elderly Cancer Patients: A Systematic Review"

_ijerph, 2022, doi:10.3390/ijerph19063424_

Round 1

Reviewer 1 Report

This is very carefully presented manuscript on psychological effects of cancer in elderly patients. Despite huge material, different patient’s populations and various definitions of the problems the Authors presented this material in clear way, easy to read, and  with great interest.

The main limitation of the study, in reviewer’s opinion, is that the findings are not new – but this problem is appropriately discussed in the text. On the other side – extensive view on the problem is very important value of this research.

There are only minor editorial mistakes: in the v. 106 (futher), 110 (considering), 396 (comma), 428 (Maeir), 450 (were), 477 (this).

Author Response

Dear Reviewer, Thank you for taking the time to consider our article and for your valuable comments and suggestions.
We have tried to respond to all recommendations. All changes in the manuscript were marked using the “Track Changes” function.

Reviewer 2 Report

That is an interesting review about distress in older patients with diagnosis of cancer. Methodology and material and methods are correct and each manuscript has been descibed in detail.

Tables are very good, and the quality of the conclusions and the discussion.

But here are many aspects which should be improved in the spelling; for example: line 169: "Blader" should be modified by "bladder";

line 279: it is necessary to write the name of the author;

line 353: "Mnini" Mental should be modified by "Mini" Mental...

Author Response

(The authors gave the same response as above.)

Reviewer 3 Report

In this systematic review, the authors investigated the clinical, sociodemographic, and psychosocial factors associated with presence of depression and anxiety in elderly with cancer. At the light of study results, apart from nonmodifiable risk factors (female sex, non-white race, low income, and low education) associated with distress, there are several modifiable ones that can be appropriately targeted and treated: pain, comorbidities, lack of social support, poor nutrition, poor functional and cognitive status have shown some or constant association with distress in elderly cancer patients. These findings support the importance of multidimensional assessment and a more comprehensive treatment of elderly patients with cancer, towards a more appropriate control of some symptoms which are sometimes not adequately targeted (e.g .pain, poor nutritional status, functional  ability).

In my opinion, the manuscript is well written in all its parts, and despite unavoidable heterogeineity of included studies, final results are important and of relevance, given the burden of depression and anxiety in cancer patients of older age. I have only some minor concerns to be addressed:

1) Line 51: anxiety and depression are common problems in older patients with cancer and without cancer. I think that an overall estimation of their prevalence (i.e separate prevalence rate of anxiety and depression) in the general older population should be provided before starting the description of such conditions in cancer older patients.

2) Line 123: the flow chart is appropriately designed. However, to enhance readibility of its description, I suggest to specify within brackets that 761 studies were identified via databases and registries, while 9 via other methods). 

3) Description of table 1 should be made more homogeneous to improve readibility, especially in the last column. I suggest to report the association between risk factors and outcomes in the same order (I think that the most clear one is that reporting first the risk factors and then the outcomes; I suggest to modify accordingly the last column of the following studies: [25], [24],  [35], [38] . Additionally, description of findings is split in a more-than-one-point list, but I think that it is sometimes confusing and should be better organized: choose whether to use a 2-point list consistently reporting factors associated with increased or decreased risk of depression/anxiety for all studies, or a 3-point list according to categorization of risk factors in the beforementioned groups (sociodemographical, psychosocial and clinical ones).

2) Some minor spell checks:

  • Use the same term to define 'sex' or 'gender' throughout the article and within tables. Similarly, use elderly instead of elder or elders, and cancer istead of "cancer disease". Additionally, Maier is often subjected of typo errors: change all "Maeir" or similar with "Maier".
  • Line 39: use such as instead of "namely".
  • Line 59: change "has" with "have". 
  • Table 1, Page 7, study of Baeza-Velasc et al: in the last column, remove the sentence "Elderly with... (NHL)" which is yet described in the population column. Then modify the following sentence with "Patients with Major Depressive Disorder (MDD) had...".
  • Table 1, page 9, study by Duc et al: in the 10th column, add "of" between Activities and Daily. In the last column, change the sentence with "...baseline, and malnutrition were...".
  • Table 1, page 9, study by Malak et al: in the last column change "was" with "were".
  • Table 1, page 10, study by Ladanineja et al: in the last column change "windowed" with "widowed".
  • Table 1, page 11, study by Canoui-Poi et al: in the 9th column decrease font size of the word symptoms; in the last one, add "with" before inpatient status. 
  • Table 1, page 14, study by Oserowsky et al: I think you should remove the sentence "Predictors of a Depression" in the last column, and write that ...-general health....urinary issues were predictors of depression. 
  • Line164: I think that it should be "17/20" instead of "17/19". 
  • Line 169: change "gastrointinal" with "gastrointestinal" and "blader" with "bladder". 
  • Line 193: change "In the other" with "In another".
  • Lines 196-198: use description of acronyms first and acronyms within brackets after, as for the abovementioned tools (see lines 187-192).
  • Line 212: add "and" after "depression,".
  • Line 239: change comma after "depressive symptoms" with semicolon.
  • Line 256: remove one full stop.
  • Line 279: add "Malak" after "investigated by" and add "that" after "et al.". 
  • Line 307: the sentence is not clear. You can write "Another authors investigated  the cancer comorbidity spectrum and found that...".
  • Line 327: use "investigated" instead of "investigate".
  • Line 353: change "Mnini" with "Mini".
  • Lines 362-463: the sentence is not clear. You can write "Similar results by Malak et al. revealed that hope was a significant predictor of depression, but not anxiety.
  • Line 365: change "as the" with "were". 
  • Line 377: add space between table and 3.
  • Lines 409-415: the sentence is not clear. You can write: "in the literature review by Maier et al, the 5 out of 16 studies carrying significant results have showed that age was associated with a greater risk of depression, and these results were congruent with those of Goldzeig et al".
  • Lines 450: change the sentence with "... and anxiety was associated with perceived..".
  • Line 495: remove "to" after "this review".

Author Response

Dear Reviwer,

Thank you for the time spent considering our paper and for the valuable comments and suggestions. We have made an attempt to address the  recommendation. Most of the changes made are related to spelling correction, but other changes were also made. Below, we present the main adjustments that have been performed, following the list of comments. All changes in the manuscript were marked using the “Track Changes” function.

1.    In the introduction, the prevalence rates of depression and anxiety in the elderly population without cancer were added (see page 2; line 50-53).
2.    The flowchart description has been slightly changed to improve its readability (see page 3; line 127-128).
3.    The last column of table 1 was changed to match the one suggested by one of the reviewers. First the risk factors were reported and finally the outcomes. The description of the findings by articles was organized into 2 points (factors associated with an increase in depression/anxiety and factors associated with a decrease in depression/anxiety) (see page 5-16).